# Substitution Arg140Gly in Hemagglutinin Reduced the Virulence of Highly Pathogenic Avian Influenza Virus H7N1

**DOI:** 10.3390/v13081584

**Published:** 2021-08-11

**Authors:** Anastasia Treshchalina, Yulia Postnikova, Elizaveta Boravleva, Alexandra Gambaryan, Alla Belyakova, Aydar Ishmukhametov, Galina Sadykova, Alexey Prilipov, Natalia Lomakina

**Affiliations:** 1Chumakov Federal Scientific Center for the Research and Development of Immune-and-Biological Products, Village of Institute of Poliomyelitis, Settlement “Moskovskiy”, 108819 Moscow, Russia; narmoriel5991@gmail.com (A.T.); elisavetbor@gmail.com (E.B.); belyakova_av@chumakovs.su (A.B.); sue_polio@chumakovs.su (A.I.); 2Biology Department, Lomonosov Moscow State University, GSP-1, Leninskie Gory, 119991 Moscow, Russia; postni.yulya@ya.ru; 3The Gamaleya National Center of Epidemiology and Microbiology of the Russian Ministry of Health, 123098 Moscow, Russia; gksadykova@gmail.com (G.S.); a_prilipov@mail.ru (A.P.); nflomakina@gmail.com (N.L.)

**Keywords:** highly pathogenic avian influenza viruses, pathogenicity factors

## Abstract

The H7 subtype of avian influenza viruses (AIV) stands out among other AIV. The H7 viruses circulate in ducks, poultry and equines and have repeatedly caused outbreaks of disease in humans. The laboratory strain A/chicken/Rostock/R0p/1934 (H7N1) (R0p), which was previously derived from the highly pathogenic strain A/FPV/Rostock/1934 (H7N1), was studied in this work to ascertain its biological property, genome stability and virulent changing mechanism. Several virus variants were obtained by serial passages in the chicken lungs. After 10 passages of this virus through the chicken lungs we obtained a much more pathogenic variant than the starting R0p. The study of intermediate passages showed a sharp increase in pathogenicity between the fifth and sixth passage. By cloning these variants, a pair of strains (R5p and R6p) was obtained, and the complete genomes of these strains were sequenced. Single amino acid substitution was revealed, namely reversion Gly140Arg in HA1. This amino acid is located at the head part of the hemagglutinin, adjacent to the receptor-binding site. In addition to the increased pathogenicity in chicken and mice, R6p differs from R5p in the shape of foci in cell culture and an increased affinity for a negatively charged receptor analogue, while maintaining a pattern of receptor-binding specificity and the pH of conformational change of HA.

## 1. Introduction

The main hosts of influenza viruses (IV) H1-H6, H8-H12 and H14 are wild ducks, while separate evolutionary branches have formed in other hosts [1]. The H13 and H16 subtypes are primarily gull viruses. The H7 subtypes and the phylogenetically close H15 also stand out among avian influenza viruses (AIV). Although they actively circulate in both ducks and poultry, evolutionary trees show that chickens are the primary hosts, and ducks are secondary hosts of H7 viruses [2]. H7 viruses in ducks are generally completely low pathogenic and lack a polybasic cleavage site in hemagglutinin (HA). However, they were easily introduced into the poultry population and rapidly evolved with increasing pathogenicity [3]. Numerous outbreaks of the H7N2 virus occurred in 1996 in Pennsylvania. Despite the absence of a polybasic cleavage site, the virus caused economic losses of the order of several million USD. Later, viruses of this evolutionary branch caused outbreaks in Miami (2001) and Virginia (2002) [4]. Outbreaks of the H7N4 virus with RKRKRG cleavage sites were recorded in Australia in 1997 [5].

More than 500 outbreaks caused by the H7N1 virus took place in Italy in 1999–2000. At first, the proteolytic site did not contain the sequence of positively charged amino acids, while by the end of this period, the presence of a “furin site” became a characteristic feature of these viruses [6]. A powerful epizootic of the H7N7 virus swept the Netherlands in 2003. This epizootic became famous, as infection with this virus was noted in 83 people, of whom one died [7].

In 2013, the first human cases of the H7N9 virus were detected in China. Since then, the laboratory has confirmed about 1600 cases, of which 615 were fatal. An intriguing circumstance was that human diseases occurred without concomitant outbreaks in poultry [8]. Over the past 6 years, H7N9 infections have occurred continuously, and the virus has accumulated mutations that change receptor specificity and increase drug resistance [9]. In 2017, a variant of the H7N9 virus with a polybasic cleavage site appeared [10].

H7N9 infection in humans is now under control by closing markets and vaccinating poultry in many provinces. Meanwhile, the H7N9 virus continues to shed from the environment and from wild birds, making new outbreaks likely. There were suspicions that the virus was transmitted sporadically from human to human. However, its ability to transform into a strain with regular human-to-human transmission remains unknown. The high (about 40%) mortality from the H7N9 virus remains the most dramatic factor. H7N9 vaccines for humans are currently in clinical trials. However, they are not on the market. Under these conditions, monitoring wild and migratory birds can help prevent the spread of the virus [11].

The study and tracking of mutant forms with increased pathogenicity and/or altered receptor specificity is also an important element of pre-pandemic preparation. The pathogenicity of the virus for mammals is influenced by substitutions in the PB2 and PA genes. Thus, the substitutions of Glu358Val in PB2 and Pro190Ser and Gln400Pro in PA attenuate the virus for mice and reduce the inflammatory process after infection [12]. It has been shown that substitutions at positions 111, 146 and 340 of hemagglutinin H7 increase the pathogenicity and transmissibility of the virus [13]. A single substitution of Gly228Ser in H7 HA affects the tissue tropism of the virus and dramatically increases the affinity for human tracheal cells [10].

Chicken plague virus A/FPV/Rostock/34 (H7N1) is a highly pathogenic avian influenza virus (HPAIV) that caused outbreaks in chickens in the 1930s. The main determinant of the high pathogenicity of HPAIV is the polybasic cleavage site of HA, which is cleaved by furins (intracellular serine proteases of animal cells). This type of cleavage allows the virus to multiply in the internal organs and leads to fatal systemic infection, in contrast to local infection caused by low pathogenic influenza viruses with monobasic cleavage sites. The study of the spread of the A/FPV/Rostock/34 virus in the host organism and tissue tropism showed that the virus caused a generalized infection, which was strictly limited to endothelial cells in all organs. Endotheliotropism is determined, on the one hand, by the cleavability of HA by furins, which ensures the penetration of the virus into the vascular system, and, on the other hand, by the polarity of virion budding and the pattern of receptor expression, which prevents the spread of infection into the tissues surrounding the endothelium [14].

The strong dependence of the pathogenicity on the structure of the cleavage site make it possible to construct attenuated variants with a shortened or altered cleavage site, which can serve as live vaccines or producers of inactivated vaccines against avian influenza viruses of the H7 subtype [15].

Another way to attenuate H7 viruses is to obtain temperature-sensitive mutants that multiply only at low temperatures. Corresponding mutations are usually located in one of three regions of the HA: (i) at the end of the alpha helix, (ii) between the globular region and the stem and (iii) in the basal stem domain. These regions are critical for HA assembly and hence transport, such that mutants cannot leave the endoplasmic reticulum at non-permissive temperatures and are incorrectly trimerized [16,17]. The Ile512Leu mutation in PB2 also led to temperature sensitivity and attenuation [18].

The H7 viruses differ from other AIVs not only in their epidemiology, but also in the structure of the receptor-binding site (RBS). Pro185, which is conservative for all subtypes, is replaced by the Ser in H7 HA.

The residues at positions 185–189 of H7 HA have small side chains (usually Ser-Gly-Ser-Thr-Thr), while other avian viruses besides Pro185 have at least one large amino acid residue at these positions. As a rule, Lys or Arg are located in positions 193 of the RBS in H7 HA, which is also typical in chicken H5N1 viruses. The positively charged amino acid residue 193 provides binding of the negatively charged sulfo group of the Neu5Acα2-3Galβ1-4-(6-O-Su)GlcNAc (Su3’SLN) and Neu5Acα2-3Galβ1-4(Fucα1-3)(6-Su)GlcNAc (SuSLe^x^) receptors [2]. Another characteristic property of H7 HA is a pair of arginines at positions 140 and 141. The Arg141 is strongly conservative for H7 HA, while the Arg140 in rare cases is replaced by Lys or a small amino acid.

The study of the causal relationship between the structural features of the RBS and the biology of H7 viruses is the goal of this work. The laboratory strain A/chicken/Rostock/R0p/1934(H7N1) (R0p) was passed through the lungs of chickens, and passage variants were compared with the parental virus. The mechanism of increasing pathogenicity was investigated.

## 2. Materials and Methods

### 2.1. Reagents

Fetuin and horseradish peroxidase were from Serva, Switzerland. Antibodies against mouse and chicken immunoglobulins conjugated with horseradish peroxidase were from Sigma-Aldrich, Saint Louis, MA, USA. Sialylglycopolymers were from GlycoNZ (Auckland, New Zealand). MDCK cells, American Type Culture Collection number ATCC CCL-34, were kindly provided by Mikhail Matrosovich at the Institute of Virology, Germany.

### 2.2. Viruses

The laboratory strain A/chicken/Rostock/R0p/1934(H7N1) (R0p), derived from A/FPV/Rostock/1934 (H7N1), was obtained from the collection of the Chumakov Federal Scientific Center. It was cloned, sequenced and used for further work. Low pathogenic mallard virus A/mallard Sweden/91/02 (H7N9) was kindly provided by Dr. R. Fouchier (Department of Virology, Erasmus Medical Center, the Netherlands). Passage variants of R0p were obtained in this study (see Section 2.7). Ten-day-old embryonated chicken eggs (CE) were inoculated with 10^2^ EID_50_ of viruses, incubated at 36 °C, monitored and cooled immediately after death or after 60 h of incubation. Infectious allantoic fluids (IAF) were harvested and tested by hemagglutination assay. The virus amount was expressed in hemagglutinating units (HAU). The 50% infective dose (EID_50_) for each virus stock was determined by titration in CE.

### 2.3. Animals

Chickens and embryonated chicken eggs were purchased from state poultry farm “Ptichnoe” (Moscow, Russia). All studies with HPAIV viruses were conducted in a biosafety level 3 containment facility. BALB/c mice (weight in the range of 8 to 10 g) were purchased from “Lesnoye” farm, Moscow, Russia.

### 2.4. Ethics Statement

Studies involving animals were performed in accordance with the European Convention for the Protection of Vertebrate Animals used for Experimental and Other Scientific Purposes, Strasbourg, 18 March 1986. All appropriate measures were taken to ameliorate animal suffering. In total, 94 chickens were used in the study; 39 chickens survived and were subsequently kept in the bird housing facility for repeated detection of antibody levels, and 55 chickens were humanely euthanized after they showed signs of severe disease. The study design was approved by the Ethics Committee of the Chumakov Federal scientific center, Moscow, Russia (Approval #4 from 2 December 2014).

### 2.5. Sequencing

Viral RNA was isolated from the allantoic fluid of infected chicken embryos with a commercial QIAamp Viral RNA mini kit (# 52904, Qiagen, Hilden, Germany). Full-length viral genome segments were obtained by reverse transcription and PCR with specific terminal primers [19], MMLV and Taq-polymerase (Alpha-Ferment Ltd., Moscow, Russia). The amplified fragments were separated by electrophoresis in 1–1.3% agarose gel and subsequently extracted from the gel with the Diatom DNA Elution kit (Isogene Laboratory Ltd., Moscow, Russia, # D1031). Sequencing reactions were performed with terminal or internal primers [20] with the BrightDye ™ Terminator Cycle Sequencing Kit v3.1 (NimaGen, the Netherlands), followed by analysis on an ABI PRISM 3100-Avant Genetic Analyser (Applied Biosystems, Foster City, CA, USA). The Lasergene software package (DNASTAR Inc., Madison, WI, USA) was used for assembly and analysis of nucleotide sequences. The complete genomes were sequenced for the original strain and its variant obtained after 10 passages in chicken lungs, as well as cloned variants of R5p and R6p. Their GenBank Accession Numbers are MT914267-MT914274, MT916934-MT916941, MT916987-MT916994 and MT917013-MT917020.

### 2.6. Infection of Chickens

The 7-day-old or 56-day-old chickens were infected intranasally with undiluted allantoic fluid containing the 10^6^ EID_50_ of tested viruses. All birds were assessed daily for body weight, clinical signs of disease and mortality.

### 2.7. Passaging the Virus through Chicken Lungs

The virus was passaged through the lungs of chickens, by infecting subsequent chickens with supernatant of a chicken lung homogenate from the previous passage. For the first passage, four 7-day-old chickens were infected intranasally with undiluted virus-containing IAF (100 μL IAF containing the 10^6^ EID50 of R0p per chicken). On day 3 after infection, one chick was humanely euthanized; lung was excised, homogenized, and centrifuged. The rest of the chicks were monitored for signs of disease. The supernatant of lung homogenate was aliquoted and frozen. One aliquot was used for infection of the new group of chickens. Lung homogenates of passage variants R5p, R6p and R10p were cloned in chicken embryos, sequenced and aliquoted. These stocks were used for retesting in chickens, mice and cell cultures, for determination of receptor specificity, and for other purposes.

### 2.8. Assessment of Intravenous Pathogenicity

The intravenous pathogenicity was measured out according to the World Organization for Animal Health recommendation [21] with minor modifications. Six-week-old specific pathogen-free chickens (SPF) with no previous history of vaccination against influenza virus were used. Each group contained three chickens. Chickens were injected intravenously into a wing vein, using 0.2 mL of an inoculum containing 1:10 dilution (using sterile PBS) of IAF containing the 10^5^ EID_50_ of viruses. Birds were examined twice daily for up to 10 days. At each observation, each bird was scored 0 if normal, 1 if sick, 2 if severely sick and 3 if dead. If birds were too sick to eat or drink, they were humanely sacrificed and scored as dead at the next observation. Dead birds were assigned a score of 3, up to the tenth day of the experiment. The scores for all birds in the group were summed up over all 10 days of observation. The result was divided by the product of the number of birds in the group by the number of observations. The maximum possible index value calculated using this method is 3.0.

### 2.9. Determination of the pH Value of the HA Conformational Change (Hemolysis Test)

Allantoic fluid clarified by low-speed centrifugation was diluted with PBS to 128 HAU. To 250 μL of the obtained viral sample, 50 μL of 2.5% chicken erythrocytes diluted in the same buffer was added and incubated, periodically shaking, at +4 °C for 1 h. The erythrocytes with the virus adsorbed on them were centrifuged at 2800 rpm/min for 1 min at +4 °C, the supernatant was removed and 250 μL of 0.1 M MES buffer with a pH in the range of 4.5 to 6.0 was added. After that, samples were incubated with shaking for 1 h at 37 °C. Untreated erythrocytes without virus served as a negative control, and erythrocytes with the addition of 0.5% Tween-20 served as a positive control. After incubation, the samples were centrifuged for 1 min at 2800 rpm, and 170 μL of the supernatant was transferred to a flat-bottom 96-well plate for measuring optical density at a wavelength of 415 nm using an iMark Microplate Reader (BioRad, Hercules, CA, USA). Based on the measurement results, a graph was constructed on which the value of the pH-dependent conformational change of HA was determined.

### 2.10. Determination of Virus Affinity with the Receptor Analogue—Fetuin

The viruses’ affinity for the peroxidase-labeled fetuin was determined in a direct solid-phase binding assay [22]. In brief, 100 μL of IAFs were added to each well of the fetuin-coated microplates. After incubation at 4 °C overnight, the plates were washed with ice-cold washing buffer (0.02% Tween 80 in PBS). Serial twofold dilutions of peroxidase-labeled fetuin in the reaction buffer (RB; 0.02% Tween 80, 0.02% bovine serum albumin, 1 μM oseltamivir carboxylate in PBS) were added into the wells (50 μL/well), and the plates were incubated at 4 °C for 1 h. After washing, the peroxidase activity in the wells was assayed with tetramethylbenzidine substrate solution. The absorbencies at 450 nm were determined, and the data were converted to Scatchard plots (A_450_/C versus A_450_), where C is the fetuin concentration expressed in μM sialic acid.

### 2.11. Determination of Receptor Specificity by Competitive Inhibition

The association constants of viral complexes with non-labeled sialylglycopolymers (SGPs) were determined in a solid-phase fetuin-binding inhibition assay [23]. The viruses were adsorbed in the well of fetuin-coated plates as described above. Serial twofold dilutions of SGPs in solution of peroxidase-labeled fetuin in RB were added into the wells (50 μL/well), and the plates were incubated at 4 °C for 1 h. After washing, the peroxidase activity in the wells was assayed as described above. The absorbencies at 450 nm were measured, transferred to a PC and processed using Microsoft Excel software.

### 2.12. Infection of Mice

Six-week-old BALB/c mice were used. From 3 up to 5 groups of six mice were formed for every virus tested. Each group contained mice infected with the same dose of virus. Groups of mice were anesthetized and inoculated intranasally with placebo (PBS) or dilutions of IAF with virus doses specified in Section 3.5. Survival and body weight following infection were monitored daily. On day 15 post infection, serum samples were taken from survivor mice for antibody titration.

### 2.13. Measurement of Antibodies against Influenza Viruses in Mouse Sera

The levels of antibody were assessed by ELISA with anti-mouse IgG. Nunc MaxiSorp plates (Thermo Fisher Scientific Inc., Waltham, MA, USA) were coated with fetuin. A total of 100 μL of IAF was added to columns A-G of fetuin-coated microplates. Allantoic fluid of uninfected embryos was added to column H. These wells served as a control for non-specific serum binding. The plate was incubated overnight at 4 °C, then washed and blocked with 0.2% BSA solution in PBS, 1 h. The blocking solution was removed, and 100 μL of buffer (0.1% Tween-20, 0.2% BSA on PBS) was added to the wells, on which the sera were titrated, starting from a dilution of 1:20. Serum at a dilution of 1/20 was added to the wells of column H. Sera from uninfected mice served as negative controls. Incubation was performed for 4 h at 4 °C. After washing, peroxidase-labeled antibodies against mouse immunoglobulins (Sigma) were added. Incubation was performed for 2 h; after washing, a color reaction with tetramethylbenzidine substrate solution was carried out.

### 2.14. Foci Formation after Infection of MDCK Cells

The MDCK cells (ATCC CCL-34) were grown in 96-well plates. The cultures were washed, and 200 μL of the Gibco DMEM (Thermo Fisher Scientific Inc., Grand Island, NY, USA) medium with 0.1% BSA were added to the wells. A total of 50 μL of IAFs was added to the outermost wells and titrated by transferring 50 μL of the solution. After 16 h, glutaraldehyde solution (20 μL) was added to the wells to a final concentration of 0.02%, incubated for 30 min; the medium was poured out, and the wells were washed. Solution of chicken antibodies to A/FPV/Rostock/34 in PBS supplemented with 0.1% Tween-20, 0.2% BSA was added, and the plates were incubated for 2 h at 4 °C and washed. Solution of anti-chicken antibodies conjugated with horseradish peroxidase was added, and the plates were incubated for 1 h at 4 °C and washed. Infected cells were visualized by incubation with 0.1 mL of substrate solution per well (0.05% 3-amino-9-ethylcarbazole, 0.01% H_2_O_2_ in 0.05 M sodium acetate buffer, pH 5.5) for 30 min.

### 2.15. Statistical Analysis

Statistical significance was determined with GraphPad Prism version 7 (Graphpad Software, Inc. San Diego, CA, USA) using the Student’s *t*-test. Statistical significance was defined as *p*  <  0.05.

### 2.16. Molecular Models

Atomic coordinates of H7 HA (1ti8) were obtained from the Protein Data Bank [24]. The molecular model was generated with DS ViewerPro 5.0 software (Version: 5.0. File name: DSViewerPro.exe. Accelrys Inc. San Diego, CA, USA).

## 3. Results

### 3.1. Emergence of Mutations during Passaging of the H7N1 Virus through Chicken Lungs

The laboratory strain A/chicken/Rostock/R0p/1934 (R0p) was studied to find its genome stability and the pathogenicity factors of H7N1 viruses. Ten serial passages of the strain R0p were carried out through chicken lungs and several virus variants were obtained that differed in virulence. The complete genome sequencing of the final tenth-passage variant (R10p) revealed nine amino acid (a.a.) substitutions compared to the initial variant (R0p), namely Val109Phe in PB2, Gln621Lys in PB1, Thr32Ala and Leu586Phe in PA, Gly140Arg in HA1 and Ala101Thr in HA2 (numbering by H3), Ser82Arg in M2 and Arg118Lys and Met124Arg in NS1 (Table 1). No differences were found in proteins NA, NP, M1 and NS2. Among these mutations, the Gly140Arg substitution in HA1 is a reversal, since Arg is located at position 140 of HA1 in all derivate strains of the A/FPV/Rostock/34 (H7N1) according to the GenBank data (V01105, GU052946, CY077420, M24457).

### 3.2. Arg140 Arrangement in the HA Molecule

A pair of arginines in positions 140 and 141 is a characteristic property of H7 HA. Arg141 in HA1 is a conservative amino acid in viruses of some subtypes. Arg140 is unique to the H7 subtype, and it is very conservative. Arg140 is a neighbor of the constant Cys139, which forms a disulfide bridge with the constant Cys97, which, in turn, is a neighbor of the key amino acid of receptor-binding site Tyr98. Arg140 is located at the edge of the RBS, at the apex of the hemagglutinin (Figure 1). The positively charged atoms of the Arg140 amino groups are directly adjacent to the edge of the RBS and can influence binding to the negatively charged sialic acid, the terminal group of the receptor.

### 3.3. Increasing the Pathogenicity of Virus Variants during Passage through Chicken Lungs

The original laboratory strain R0p was low pathogenic in chickens. Chicken embryos infected with this virus died within 24–48 h, nevertheless, infected chicks, even one weeks of age, died rarely (Table 2). During the passage of this virus through the chicken lungs until the fifth passage, no changes in pathogenicity occurred, but starting from the sixth passage the virus became very pathogenic in chickens. None of the birds infected with variants obtained after the fifth passage survived (Table 2).

Since a sharp jump in the pathogenicity occurs between the fifth and sixth passage, we carried out a complete sequencing of these variants (R5p and R6p). Among all viral proteins of R5p and R6p, only one amino acid difference was found: replacement Gly140Arg in HA1 (Table 1).

### 3.4. Intravenous Pathogenicity of Strains R5p and R6p

Two groups of three 6-week-old chickens were infected intravenously with 10-times-diluted IAFs containing the 10^5^ EID_50_ of the R5p and the R6p strains. One bird treated with the R5p strain died on the second day, the second showed signs of disease until the 10th day, and the third bird was sick until the fourth day and recovered. All birds infected with the R6p strain died on day 2 after infection. The pathogenicity index for these strains was 1.7 and 2.8, respectively. According to the OIE instructions, avian influenza viruses with a pathogenicity index of 1.2 or more (maximum value 3.0) are considered highly virulent. Thus, the R6p is a typical highly virulent virus, while the R5p retains its highly virulent status, although its pathogenicity was reduced in comparison with the R6p.

### 3.5. Pathogenicity of R5p and R6p for Mice, Compared with Highly Pathogenic Virus A/Chicken/Kurgan/5/2005 (H5N1) and Low Pathogenic Virus A/Mallard/Sweden/91/02(H7N9)

Mice were infected intranasally with varying doses (10^1^–10^5^ EID/mouse) of R5p and R6p strains, as well as low pathogenic virus m/Sw/91/02 (10^4^–10^6^ EID/mouse) and highly pathogenic A/chicken/Kurgan/5/2005 (H5N1) (10^−1^–10^3^ EID/mouse), which were taken for comparison. The mice were monitored and weighed daily. We recorded three variants of response to virus infection in mice: (1) immune response without signs of disease, (2) illness with subsequent recovery and (3) death. Mice that lost weight within 5 days and survived were classified as diseased. On the 15th day, antibodies against homologous virus were measured in all surviving mice. Figure 2 shows an example of antibodies level determination for lowest dose of R5p.

The high doses of strain R5p as well as R6p killed mice. The lethal dose of the R6p strain was an order of magnitude lower than that of the R5p strain (Table 3, Figure 3). Infections with sublethal doses of both strains allowed us to observe a reliable picture of the disease in mice. The mallard/Sweden/91/02 (H7N9) virus caused an asymptomatic infection in mice when challenged at the maximum dose, 10^6^ EID_50_, and elicited an immune response at the lowest dose, 10^4^ EID_50_. The lethal dose of H5N1 virus differs little from its immunogenic dose; only occasionally were surviving mice with antibodies detected, and diseased but survivor mice were almost never detected. Mice without signs of disease and without antibodies were only in groups inoculated with doses of 10^−1^ and 10^0^ EID_50_; obviously, these mice were not infected.

The second column shows the aa sequence of the cleavage site. ImD_50_ is the decimal logarithm of the virus dose, expressed in EID_50_, leading to an immune response in 50% of mice. DD_50_ is the logarithm of the virus dose leading to disease in 50% of mice. LD_50_ is the logarithm of the virus dose resulting in the death of 50% of the mice. The averaged data of three experiments are given.

The pathogenicity of the studied viruses correlates with the aa sequence of the cleavage site of HA, although in highly pathogenic H7 and H5N1 viruses it may be due to other factors associated with other genes [25]. Since the only difference between R5p and R6p is the substitution of Gly140Arg, the higher pathogenicity of R6p in mice can be confidently attributed to this substitution. Figure 3 shows the weight dynamics and survival of mice infected with the three above-mentioned H7 viruses in dose 10^4^ EID_50_. Of the mice infected with R5p and R6p, 1/6 and 4/6, respectively, died, while mice infected with the A/mallard/Sweden/91/02 virus suffered an asymptomatic infection. Antibodies to H7 HA were found in all surviving mice.

### 3.6. Effect of Gly140Arg Substitution on the pH-Dependent Conformational Change of HA

One of the factors influencing the pathogenicity of influenza viruses is the value of pH-dependent conformational change of HA. In viruses of wild ducks, the conformational change occurs at pH close to five. A shift in the pH transition towards a less acidic pH is characteristic of highly virulent strains of poultry influenza viruses. During attenuation of the highly pathogenic virus A/chicken/Kurgan/5/2005 (H5N1), a sharp drop in the transition pH, due to mutations in the stem part of HA, was shown [26]. It was shown that a high pH transition increases the tropism of viruses to human endothelial cells [27]. We compared the transition pH of strains R5p and R6p to test the hypothesis that substitution increases the pathogenicity of the virus by increasing the pH of transition. However, this hypothesis was not confirmed. As can be seen from Figure 4, the curves of pH-dependent erythrocyte hemolysis for strains R5p and R6p practically coincide; that is, the Gly140Arg substitution did not affect the value of the pH-dependent conformational change of the virus. Obviously, this is due to the surface arrangement of Arg140.

### 3.7. Effect of Charge Mutation Gly140Arg in Hemagglutinin on the Affinity for Receptor Analogs and on Receptor Specificity

Fetuin—fetal bovine serum albumin—is a glycoprotein that contains double- and quadruple-branched N-glycans ending in Siaα2-3Galβ1-4GlcNAc and Siaα2-6Galβ1-4GlcNAc motifs [28]. This makes it a good model receptor for most influenza viruses. We compared the affinity of R5p and R6p for fetuin labeled with horseradish peroxidase. The analysis of Fet-HRP binding to virions adsorbed on 96-well panels was performed in Scatchard coordinates (Figure 5). The affinity of fetuin for the R6p virus is two times higher than for the R5p virus (dissociation constants are 0.4 and 0.8 μM, respectively). This is probably due to the strong negative charge of fetuin and a sharp increase in the positive charge in the R6p strain compared to R5p, due to the replacement of Gly by Arg.

### 3.8. Comparison Receptor Specificity Patterns of Viruses R5p and R6p

Receptor specificity of the viruses was characterized in fetuin-binding inhibition assay using synthetic sialylglycosaccharides, attached to the polymer [29]. The structures and designations of the oligosaccharide moieties are shown below (Table 4).

An example of testing the receptor specificity patterns of viruses is shown in Figure 6. Receptor determinants 3’SL, 3’SLN, SLe^c^ and Su3’SLN reliably inhibit binding to both strains (R5p and R6p), while 6’SLN, SLe^a^ and STF do not. R5p and R6p showed similar binding profiles, typical for avian influenza viruses of the H7 subtype [2]. The similarity of the binding patterns of the R5p and R6p viruses indicates that the Gly140Arg substitution in HA1 did not affect the interaction of receptor moiety with the receptor-binding site.

### 3.9. Foci of the Infected Cells Produced by the Viruses in MDCK Monolayer

The assay used was based on the method of cultivation of influenza viruses in MDCK cells in 96-well microplates and detection of foci of infected cells by immunoperoxidase staining [30]. Infected cells were treated with chicken antibodies against R0p virus followed by peroxidase-labeled anti-chicken antibodies, and stained with aminoethylcarbazole–hydrogen peroxide solution. Cells infected with the virus acquired a red color.

Monolayers of MDCK cells were infected with R5p and R6p variants, incubated for 16 h and stained as described in the Materials and Methods section. The R5p produces large, indistinct foci, which bear gaps of noninfected cells inside the focus (Figure 7A). The foci of R6p are small, round-shaped and reveal no gaps of noninfected cells between the stained cells surrounding the center of the focus (Figure 7B). These features of the foci suggest that the R6p virus progeny spread directly from cell to cell, while R5p virus progeny are released into the culture medium and infect the distance cells. The different shape of foci produced by the R5p and R6p viruses can be explained by the higher affinity for cells of R6p than R5p.

## 4. Discussion

Viruses with H7 HA subtypes have been isolated from a wide range of hosts, such as wild ducks, poultry, horses, seals and humans. They have caused disease outbreaks in poultry (“fowl plaque”) in both hemispheres. Little genetic diversity has been observed between H7 viruses isolated from wild birds and poultry birds in the same time and region, suggesting their unrestricted interspecies transmission.

Such a wide range of hosts may be because modern H7 AIVs are descendants of viruses, which long ago had transmitted from ducks to chickens, adapted to a new host and then returned to wild ducks again, which allowed the virus to persist and spread. This is confirmed by the topology of the evolutionary trees of H7 viruses, at the base of which not duck, but chicken viruses are grouped [2].

The receptor specificity of H7 AIVs also has a dual character. They recognize the chicken receptors Su3’SLN and SuSLe^x^ as well as the main duck receptor SLe^c^. Many H7 AIVs also show some affinity for the “human” receptor, 6’SLN. Naturally, this broad receptor specificity contributes to the ability to infect a wide range of hosts. The structure of the receptor-binding site of hemagglutinin H7 indicates the rearrangement that occurred during adaptation to the new receptor in new hosts. All H7 AIVs have Lys193 in HA, similar to HPAIV H5N1. Lys193 provides increased binding of viruses to sulfated receptors Su3’SLN and SuSLe^x^ due to electrostatic interaction of the sulfate residue of the receptor with the positively charged lysine side chain [23]. The set of amino acids 185–189 located at the bottom of the RBS is unique [2]. The pair Arg140 and Arg141 is another feature of H7 HA. Arg141 in HA1 is a constant amino acid in viruses of H7, H15, H10, H14, H3 and H4 subtypes. It plays an important role in maintaining the structure of the RBS, since the Arg141 residue is inside the molecule and its positively charged groups contact with negatively charged oxygen Asp77, Gly72 and Phe147 [24]. Arg140 is unique to the H7 subtype; it is located at the edge of the receptor-binding site, at the apex of the hemagglutinin. Arg140 is conservative in H7 HA, although in rare cases it is replaced by Lys or a small amino acid.

## 5. Conclusions

Reversion of Gly140Arg dramatically increases the surface positive charge of virions and promotes the binding of the virus to the negatively charged surface and molecules. The R6p strain differs from R5p by an increased pathogenicity in chickens and mice, increased affinity for a negatively charged receptor analogue and an increased affinity for MDCK cells. At the same time, the pattern of receptor specificity and the pH of the hemagglutinin transition of the R6p do not change. The increased pathogenicity of R6p in chickens and mice is probably due to an increased affinity for negatively charged host cells and receptors.

## Figures and Tables

**Figure 1 viruses-13-01584-f001:**
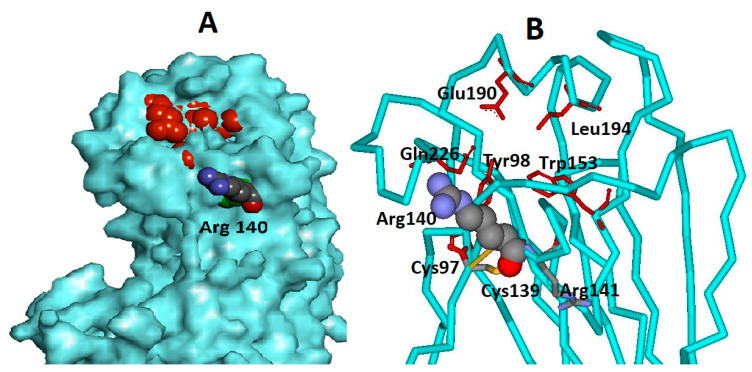
Arg140 arrangement in HA. Atomic coordinates of H7 HA (1ti8) were used [24]. (**A**) The protein surface is colored blue. The key amino acids of RBS are colored red. The Arg140 is colored by element. (**B**) Protein chain of HA near the RBS. The key amino acids of RBS are colored red and marked. The Arg140 and neighboring amino acids are colored by element.

**Figure 2 viruses-13-01584-f002:**
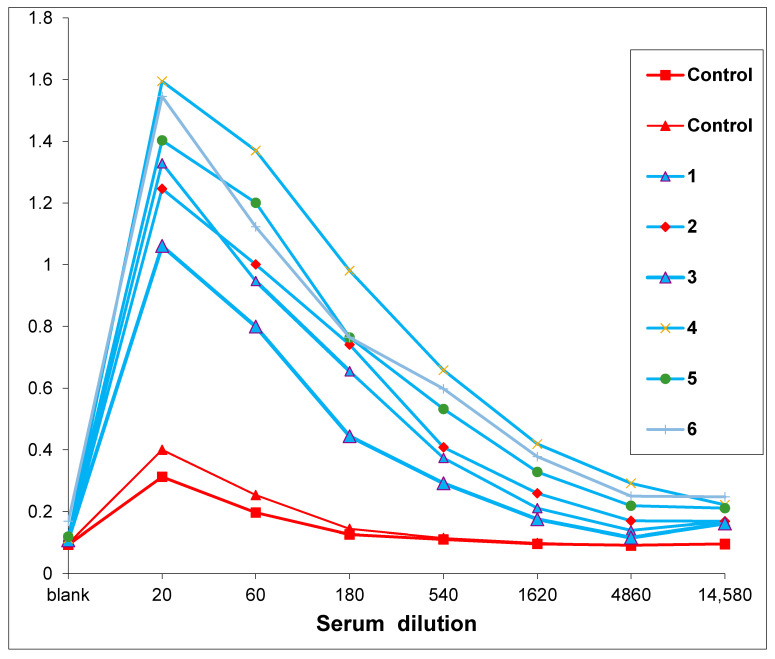
Antibodies against homologous virus measured in ELISA in serum of mice infected with 10^1^ EID_50_ of R5p. Blue curves show levels of signal for 6 infected mice. Red curves show levels of signal for 2 non-infected mice. The first point in all curves shows signals in control well (column H without virus, see Section 2.13).

**Figure 3 viruses-13-01584-f003:**
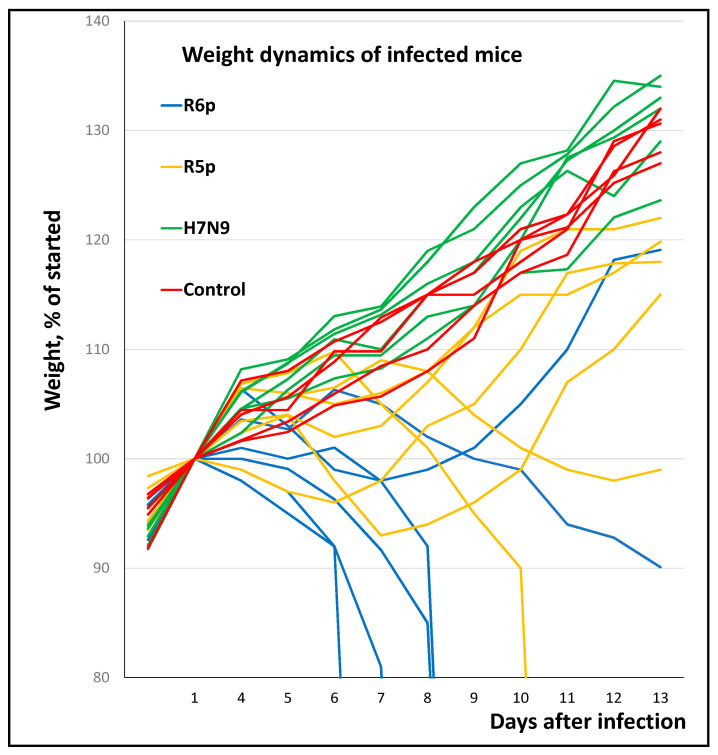
Weight dynamics and survival of mice infected with R5p, R6p and the A/mallard/Sweden/91/02 (H7N9) virus. Result of a typical experiment is presented.

**Figure 4 viruses-13-01584-f004:**
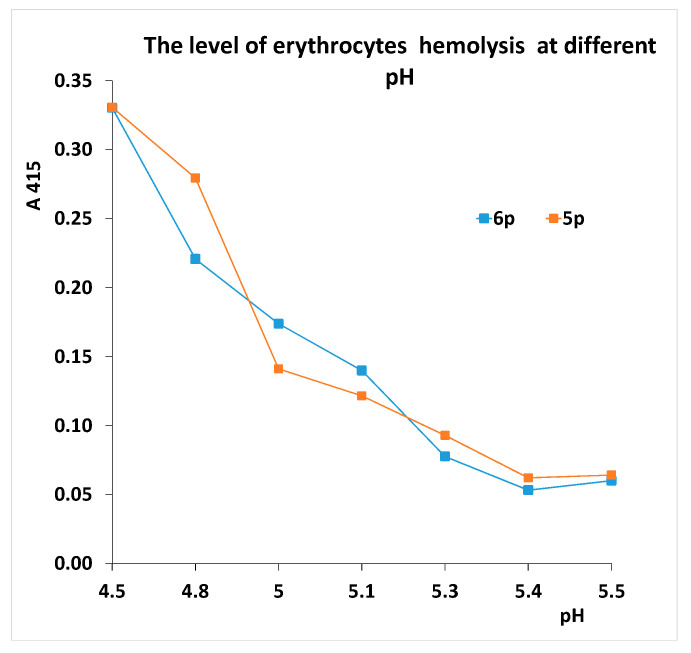
The pH dependence of erythrocyte hemolysis in complex with R5p and R6p viruses. A_415_—absorption at 415 nm, which corresponds to the degree of erythrocyte hemolysis. Result of a typical experiment is presented.

**Figure 5 viruses-13-01584-f005:**
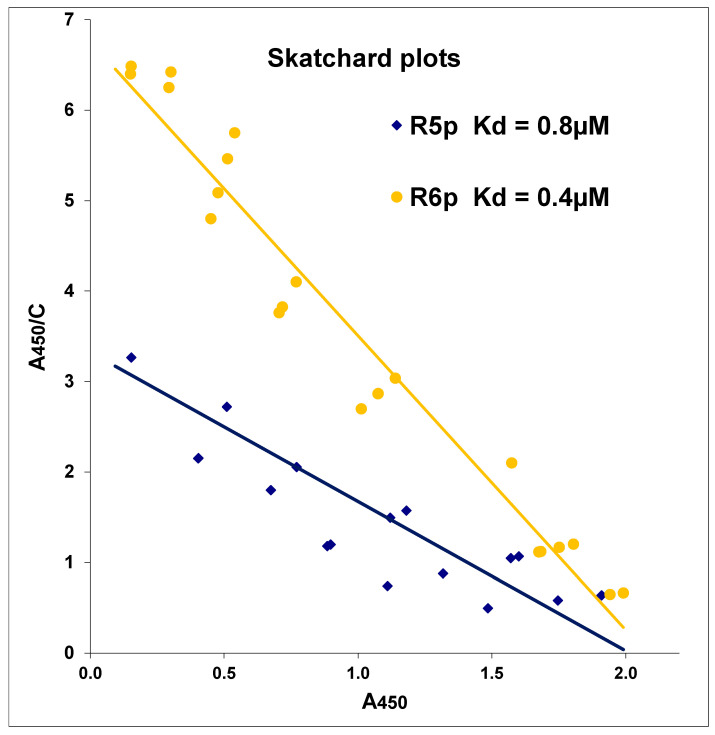
Affinity of peroxidase-labeled fetuin for virions of R5p and R6p strains. The results of testing Fet-HRP on a plate coated with R5p and R6p viruses are presented in Scatchard coordinates. The *X*-axis is the A_450_ signal reflecting the amount of bound fetuin, and the *Y*-axis is the ratio of the A_450_ signal to the fetuin concentration (expressed in µM of sialic acid) in the solution. Result of a typical experiment is presented.

**Figure 6 viruses-13-01584-f006:**
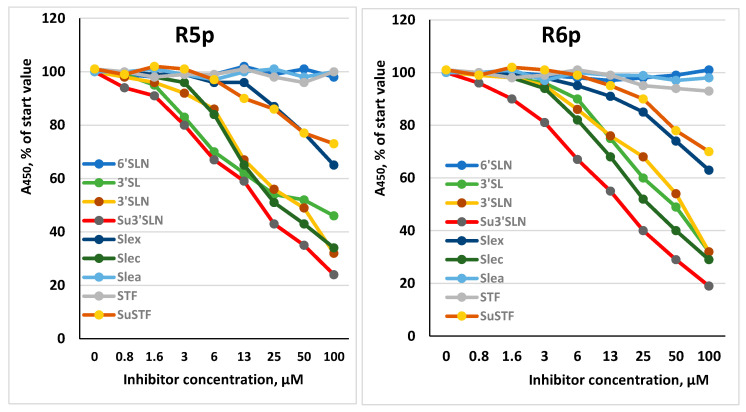
Inhibition of labeled fetuin binding to R5p and R6p viruses by the receptor analogs. Receptor analogs were added to a solution of labeled fetuin at the indicated concentrations and the level of binding of the label to the virus was measured. Result of a typical experiment is presented.

**Figure 7 viruses-13-01584-f007:**
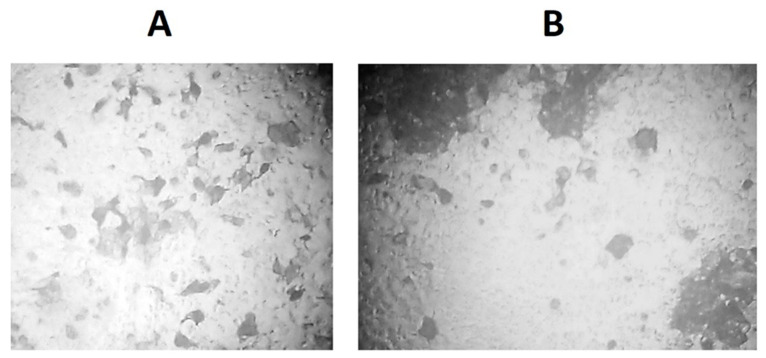
Different pictures of cells infected with R5p (**A**) and R6p (**B**) variants of influenza virus Rostock.

**Table 1 viruses-13-01584-t001:** Mutations emerging during serial passages of the R0p through chicken lungs.

Protein	PB2	PB2	PB1	PB1-F2	PA	HA1	HA2	M2	NS1	NS1
Position *	109	567	621	4	32	140	101	82	112	118
0 passages	Val	Val	Gln	Glu	Trp	Gly	Ala	Ser	Arg	Met
5 passages	Phe	Ala	Lys	Glu	Ala	Gly	Thr	Ser	Arg	Arg
6 passages	Phe	Ala	Lys	Glu	Ala	Arg	Thr	Ser	Arg	Arg
10 passages	Phe	Ala	Lys	Gly	Ala	Arg	Thr	Arg	Lys	Arg

* HA amino acids are numbered according to H3.

**Table 2 viruses-13-01584-t002:** Survival of one-week-old and eight-week-old chicks after infection with the passage variants of the virus R0p.

Chick Age		One-Week-Old Chicks		Eight-Week-Old Chicks	
Variants	Designation	Infected	Survived	Infected	Survived
m/Sw/91/02 *	H7N9	6	6	2	2
ch/Ku/5/05 *	H5N1	5	0	5	0
0 passages	R0p	4	3	5	5
1 * passage	R1p	3	1	2	2
2 passages	R2p	3	2	2	2
3 passages	R3p	3	3	6	6
4 passages	R4p	2	2	2	2
5 passages	R5p	2	2	2	2
6 passages	R6p	5	0	15	0
7 passages	R7p	2	0		
8 passages	R8p	2	0		
9 passages	R9p	2	0		
10 passages	R10p	2	0	6	0

* The data on the low pathogenic virus A/mallard/Sweden/91/02 (H7N9) (m/Sw/91/02) and the highly pathogenic virus A/chicken/Kurgan/5/2005 (H5N1) (ch/Ku/5/05) are given for comparison.

**Table 3 viruses-13-01584-t003:** Effect of Gly140Arg amino acid substitution in HA1 on pathogenicity in mice.

Virus	Cleavage Site of HA	a.a. in 140 HA1	Immunogenic Dose (ImD_50_)	Disease Dose (DD_50_)	Lethal Dose (LD_50_)
R5p	SKKRKKR|GLF	Gly	≤1	4	5
R6p	SKKRKKR|GLF	Arg	≤1	3	4
m/Sw/91/02	PKGR---|GLF	Arg	≤4	>6	>6
ch/Ku/5/05	ERRRKKR|GLF	Pro	≤0.3	0.3	0.3

**Table 4 viruses-13-01584-t004:** Sialyloligosaccharide and designation of the oligosaccharide moieties.

Sialyloligosaccharide	Designation
Neu5Acα2-6Galβ1-4GlcNAcβ-	6’SLN
Neu5Acα2-3Galβ1-4Glcβ-	3’SL
Neu5Acα2-3Galβ1-4GlcNAcβ-	3’SLN
Neu5Acα2-3Galβ1-4-(6-O-Su)GlcNAcβ-	Su3’SLN
Neu5Acα2-3Galβ1-4(Fucα1-3)GlcNAcβ-	SLe^x^
Neu5Acα2-3Galβ1-3GlcNAcβ-	SLe^c^
Neu5Acα2-3Galβ1-3(Fucα1-4)GlcNAcβ-	SLe^a^
Neu5Acα2-3Galβ1-3GalNAcα-	STF
Neu5Acα2-3Galβ1-3-(6-Su)GalNAcα-	SuSTF

## Data Availability

Not applicable.

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
