# Peer review of "Substitution Arg140Gly in Hemagglutinin Reduced the Virulence of Highly Pathogenic Avian Influenza Virus H7N1"

_viruses, 2021, doi:10.3390/v13081584_

Round 1

Reviewer 1 Report

N/A

Author Response

Following the advice of the reviewers, we rewrote the Abstract, and edited the text with the help of a person who is fluent in English.

Reviewer 2 Report

Major comments

  • The document needs to be reviewed for proper English grammar, especially the abstract section and most of the sections highlighted in yellow. There are several mistakes in the document, and some are mentioned in the minor comments. This manuscript would greatly benefit from having a native English speaker revising it or from the use of a professional editing service
  • I understand that the lung homogenates were passed into eggs before performing the analyses. I understand why this was necessary. However, did the authors performed sequence analysis straight from the lung homogenates and then from the egg passage to ensure no additional mutations appeared after the egg passage? Please clarify in the manuscript.
  • There is too much variability in the number of chickens used in this study, and the numbers a quite low to drive adequate conclusions.
  • It is unclear how many chickens were used for the IVPI. The way the results for the IVPI experiment are written (and the lack of details in the M&M section), makes me believe that only three chickens were used. If that was the case, it is unacceptable to calculate an IVPI based on those numbers. The OIE specifies that 10 chickens between 4 and 8 weeks of age are to be used. Please clarify/rephrase and provide more details in the materials and methods and the results sections.
  • How did the authors determined the immunogenic dose of virus shown in table 3? How the cut off was stablished. In addition, there is no figure show the results obtained from the ELISA. Please add a figure for the assessment of the antibody response.
  • The authors assessed the immunogenicity of the passage viruses; however, HI assays were not performed.
  • In lines 355-356, the authors strongly conclude that “Substitution Gly140Arg undoubtedly increases he pathogenicity of the virus for mice, since this is the only difference between the R5p 356 and R6p variants”. I do not completely agree as there in no effect for the H7N9 virus. It may play a role, but I would not state that “undoubtedly” increases pathogenicity. A growth kinetic analysis may be useful to determine whether both passage viruses have the same replication rate.
  • Perhaps performing a plaque assay would have been more appropriate instead of assessing foci formation through immunoperoxidase assay.

Minor comments

  • Line 2: add “of” to between “virulence and highly”
  • Edit abstract for proper English grammar. i.e. lines 20-24 contain improperly structured sentences. Line 30, use “in chickens” instead of “for chicken”.
  • Line 69: delete “to”
  • Line 105: use “H7 HA” instead of “HA H7”. Check all the manuscript and correct accordingly to maintain consistency
  • Lines 105-106: “In HA H7, the conserved for all subtypes Pro185 is replaced by Ser.” does not read properly, seems truncated, please rephrase.
  • Line 109: use “position 193” instead of “193 position”
  • Line 113: use “arginines at positions 140 and 141” instead of “arginines: 140 and 141.”
  • Line 115: delete “H7”
  • Line 116: “and the biology of H7 viruses is the goal of this work” may read better.
  • Line 117: add “the” between “in” and “HA”
  • Line 120: “with the parental virus” may read better
  • Line 130: did you mean “low pathogenic” instead of “Non-pathogenic”?
  • Line 146-148: remove the parentheses
  • Lines 146-153: the authors mentioned the number of chickens that were euthanized or died. How about the mice?
  • Line 172: please specify how much virus was given to each bird
  • Line 177: use “was” instead of “were”
  • Line 179: add “homogenate” between “lung” and “supernatant”
  • Line 180: replace “chicken” with “chickens”. Replace “was” with “were”
  • Line 183: please use “intravenous pathogenicity index”
  • Line 184: add “the” between “to” and “World”
  • Line 185: hoe many chickens were used for the IVPI?
  • Line 230: please rephrase sentence “Six group of six mice were formed for every virus tested one group for one dose of virus.”, it does not read properly.
  • Lines 230-334: Which viruses did you use to infect the mice? The passage viruses? How many groups of mice did you have total?
  • Line 233: delete the “s” from “days”
  • Line 237: replace “covered” with “coated”
  • Line 246: I suggest changing the subtitle to something like “foci formation after infection of MDCK cells”
  • Line 265: remove “the” before “chicken”
  • Line 268: add “a” between “with” and “laboratory”
  • Line 269: the word “relatively” does not seem to be the appropriate word to be used here; perhaps “compared to” is more appropriate based on the context.
  • Line 282: delete “the virus”
  • Line 283: delete “the”
  • Lines 286-288: highlight the aa positions mentioned in these lines on figure 1
  • Line 288: perhaps you meant “Arg140” instead of “Arg141”
  • Line 299: delete “the” before “chicken”. Also, please keep consistency throughout the manuscript. Either use “R0p” or use “A/chicken/Rostock/R0p/1934” but choose one to avoid confusing the readers
  • Line 301: use “in chickens” instead of “for chickens”
  • Line 302: here the authors mention that the embryos infected with the virus carrying Gly140 died within 24-48 hpi. How come was the virus less pathogenic in embryos as mentioned in lines 280-281? Was the pathogenicity assessed by any other means? Please clarify.
  • Line 311: the proper term to be used is “low pathogenic” instead of “apathogenic”, please correct accordingly throughout the manuscript as “apathogenic” appears in multiple sections
  • Line 314-315: this was already mentioned in section 3.1. the way it is mentioned here makes the reader believe the sequencing happened after the pathogenicity study in chickens. Please clarify, rephrase or remove
  • Line 316: please use “intravenous pathogenicity index”
  • Line 317: use “intravenously”
  • Line 333: “Mice that lost weight within 5 days and survived…” may read better than on its present form. Please rephrase this sentence
  • Line 334: rephrase sentence about the antibody measurement, it does not read properly. Perhaps “antibodies against homologous virus” may read better.
  • Line 338: please replace “structure” with “aa sequence”. Same for line 353
  • Table 3: how come the DD50 and the LD50 for the H7N9 is >6 when the mice were inoculated with doses ranging from 10^1 through 10^5? Please clarify.
  • Line 343: delete the last “s” from “survivors”
  • Line 424: add an ‘s” at the endo of “Monolayer” and replace “was” with “were”

Author Response

This manuscript is a resubmission of an earlier submission. The following is a list of the peer review reports and author responses from that submission.

Round 1

Reviewer 1 Report

Serial passaging of a clone of HP(?)AIV Rostock/34 (H7N1) in chicken lungs in vivo brought up several AA substitutions in different viral proteins including HA. A sharp gain in virulence for infections of chickens and mice was noted between passages 5 and 6 associated with a G140R mutation in HA. R140 increased virulence, increased affinity to fetuin and altered focus morphology in MDCK cells but did not shift receptor specificity and fusion pH. The authors vaguely conclude that increased affinity for negatively charged host cells and receptors is probably responsible for increased pathogenicity.

The study provides original and new information of interest for specialists in the AIV field. The manuscript, however, suffers from imprecisions in study design, presentation and interpretation of results. The conclusions remain vague and are not always supported by the data shown. Part of the problems may be due to linguistic shortcomings in style and grammar.

Major concerns arise with the following:

  1. Potential violation of the DURC guidelines: Serial in vivo passaging of a pathogen aiming at an increase of pathogenicity clearly violates these guidelines (gain of function). Therefore, the authors must reliably prove that isolates/strains/lineages of the virus they were studying have existed in nature before they started their experiments. The authors state that this mutation is a reversion. However, why has their isolate of Rostock/34 a G at position 140, and do other isolates of that virus stored elsewhere have similar mutations at this site? The passage history of this virus needs to be fully disclosed.
  2. Imprecisions in descriptions and phrases:

- The status of pathogenicity of the original virus for chickens and mice remains unclear (at least to this reviewer). Authors compare R5 and R6 but not R0.

- Type of 140 substitution: Is it R140G (title) or G140R (elsewhere in the text).

- Chicken as original hosts of H7 viruses: Could this impression be biased by the fact that the oldest H7 sequences available to date stem from chicken viruses? A nadir seems to have existed somewhere in the not so distant past of H7 viruses which may have complicated the evolution of this subtype (Worobey et al., 2014, PMID: 24531761)?

- Increased affinity of R6 to MDCK cells: Not clear on what observations this is based. The foci study likely does not justify this conclusion; such experiments would examine cell-to-cell spread and cleavability of the HACS in the presence/absence of specific proteases? Such studies should be backed up by plaque morphology descriptions which allow another view on c2c spread.

- “maintaining a receptor profile”, Abstract: Which receptor profile is meant?

- “increased affinity for negatively charged host cells and receptors”, Discussion: Which receptors are meant here? No differences were seen when comparing R5 and R6?

- Various typos, misspellings, use of phrases etc. e.g., conservative – conserved, reversal – reversion, PBS – RBS, suffered asymptomatic infection – no suffering when there are no symptoms, bi- and dendritic N-glycans – bi-antennary and…

- Figure 6: in A there are only single infected cells visible, while in B foci seem to be visible.

Reviewer 2 Report

The manuscript by Treshchalina et al seeks to understand the mechanisms of increased virulence on a viral mutation level as the H7 viruses are passaged through birds and whether increased infectiousness to mice (surrogate for humans) occurs with these mutations. A few issues were found:

Line 302: One of birds....english as stated is awkward, please revise

Figure 3: Axis on both X and Y are not centered. Use decimal and not commas. Should there be error bars?

Figure 5: Y axis should be centered

Figure 2: if each group is taken as a mean, are they statistically different? X axis is also not centered

Figure 6 images are not very clear. Are there better pictures than those?

LIne 317: What do you mean the surviving mice were classified diseased?

Line 318: Did you mean antibodies were found? "Determined" is not a good word choice

Line 416: should that be part of the figure label for fig 6? It appears separate from it.

Round 2

Reviewer 1 Report

Overall I regret to state that the manuscript although it has improved it is still does not warrant publication in its current state. Extensive editing is still required.

L20 Which conditions? Pls give a reference here or explain.

L24 "hundreds of times", pls be specific

L25 What are intermediate passages?

L33/34, if the receptor binding properties changed, the receptor specificity needs to change also? These seem to be contradictory statements?

Introduction: The authors list all major HP H7 outbreaks but the connectivity to their study is not clear.

L57: Holland and the Netherlands are the same country; do you mean Germany here?

L66/67: Pls add a reference for this statement and rephrase "shedding from the environment".

L158: Pls describe how the strain was attenuated and why or give a reference.

L208: What is measured at 415 nm, which substance?

L282: Location of R140 in the HA molecule

L283 and elsewhere: "constant", "conservative" vs conserved; pls change.

L289-292: The authors do not show data to support this assumption. Either they present the data or move this as a speculation to the discussion. Indicate the coloring in figure 1 in brackets in this paragraph and explain here and in the legend to figure 1 what the blue and black coloring stands for.

L299: It may be easier to speak of the HA G140 as a position.

L301: "died rarely". The classification of the G140 Rostock-at remains unclear. Is it LP or HP; if LP why were deaths observed? Is the attenuation incomplete? Maybe other mutations play a role also? In this respect, the IVPI described in L315 and following is incomplete. Instead of three, ten chickens must be used according to the OIE standard. The interpretation of the data now indicates that both R5p (IVPI 1.7) and R6p (IVPI 2.8) are HPAIV and thus, G140R does not provide the switch in pathogenicity as the authors claim. "Attenuation" would mean reduction of pathogenicity from HP to LP status; now it is a "reduction in virulence". In lines 323/4 the authors rephrase their statement which is more appropriately fitting the data shown, but it still remains unclear what they state in the abstract (L24).

L335: In addition to showing the doses in Table 3, also the absolute numbers of mice used/died should be shown.

Figure 6 is not of sufficient quality to distinguish the two types of plaques induced by the different passage variants.

The authors have produced original and potentially interesting data. However, their presentation and interpretation is still not sufficient for publication.